# Insights into Localized Crystallization in the 3D-Cone Solar Evaporator for High-Salinity Desalination

**DOI:** 10.3390/ma18112610

**Published:** 2025-06-03

**Authors:** Ruolan Tang, Wanqi Chen, Bo Yang, Banghe Lv, Haile Yan, Song Li, Liang Zuo

**Affiliations:** Key Laboratory for Anisotropy and Texture of Materials, School of Material Science and Engineering, Northeastern University, Shenyang 110819, China; t1434691930@163.com (R.T.); cwqq0824@163.com (W.C.); 20213865@stu.neu.edu.cn (B.L.); yanhaile@mail.neu.edu.cn (H.Y.); lis@atm.neu.edu.cn (S.L.); lzuo@mail.neu.edu.cn (L.Z.)

**Keywords:** 3D-cone evaporator, solar desalination, salt precipitation, photothermal conversion

## Abstract

Solar-driven interfacial evaporation desalination is regarded as a promising solution to address freshwater scarcity. However, salt deposition remains a significant challenge. While structural designs such as designated deposition sites can control crystallization, the mechanisms of salt precipitation at specific locations are still unclear. In the present work, we designed a three-dimensional conical evaporator using low-cost cellulose paper for efficient solar-driven desalination. This innovative evaporator design achieves controlled salt crystallization by meticulously balancing the rates of salt diffusion and accumulation, thereby directing salt precipitation to a predetermined location approximately 1.4 cm above the conical base. This phenomenon arises from temperature variations across the evaporator’s three-dimensional surface, which induce differences in water surface tension and create favorable sites for salt precipitation. Such a salt management strategy allows for continuous operation for up to 8 h in high-salinity conditions (24.5 wt.%) without compromising performance. Under one sun irradiation, the evaporator demonstrates exceptional performance, with an evaporation rate of 2.54 kg·m^−2^·h^−1^ and an impressive energy conversion efficiency of 93.7%. This approach provides valuable insights into the salt precipitation mechanism, contributing to the future design of three-dimensional evaporators and innovative salt collection strategies.

## 1. Introduction

The intensifying global freshwater crisis, driven by concurrent population growth and industrial expansion, has elevated freshwater security to a critical priority in sustainable development agendas [1,2,3]. Paradoxically, while terrestrial water resources are being depleted at unprecedented rates, over 97% of planetary water remains locked in saline reservoirs [4,5,6]. This hydrological imbalance has propelled advanced desalination technologies to the forefront of water security solutions, particularly those leveraging renewable energy inputs. Among emerging innovations, solar-driven interfacial water evaporation systems [7,8,9,10] demonstrate exceptional promise through their synergistic combination of high photothermal conversion efficiency and minimal carbon footprint. Their operational mechanism, utilizing plasmonic nanoabsorbers (e.g., gold or aluminum nanoparticles) [11,12,13,14] or bio-inspired structures inspired by natural water transport mechanisms [15,16] to localize thermal energy at liquid–air interfaces, enables desalination performance exceeding conventional reverse osmosis systems in energy-constrained environments [17,18]. Nevertheless, practical implementation reveals a critical operational challenge. It is the localized vapor generation at the evaporation interface that induces progressive salt accumulation on the evaporator surface. This crystallization phenomenon progressively diminishes photon absorption capacity, accelerates material degradation through salting-induced stress, and ultimately necessitates system shutdown for maintenance [19,20]. Consequently, developing robust salt management protocols has become paramount for achieving dual objectives, sustaining uninterrupted vapor production while enabling concurrent mineral recovery through controlled crystallization [21,22].

Current desalination strategies are broadly classified into three categories: separation, dilution, and salt crystallization [19]. Separation techniques employ physical isolation between the evaporation and brine layers. Representative approaches include Janus structures with spatially modulated hydrophilicity/hydrophobicity, such as implemented using composite membranes [23,24], PDMS-modified MF foam [25], and aerogel [26]; and Donnan effect-driven ion exclusion using polyelectrolyte hydrogels, such as adding sodium polyacrylate (PAA) [27,28] and amine-functionalized hydrogels [29], both effectively suppressing salt ion diffusion to the evaporation interface. Dilution strategies rely on engineered fluidic architectures such as wood-derived evaporators [30] or directional freeze-casted hydrogels [31,32], featuring low-tortuosity vertical/lateral channels that rapidly transport concentrated brine to bulk water reservoirs through interconnected pores, thereby delaying salt saturation. The evaporation rate of two-dimensional (2D) photothermal materials [26,33] typically ranges from 1.5 to 2.0 kg·m−^2^·h−^1^, while three-dimensional (3D) structured materials [10,34] generally achieve higher rates between 2.0 and 6.0 kg·m−^2^·h−^1^. Although certain 2D and 3D materials have been engineered with tailored surface or bulk structures to induce preferential salt crystallization at specific regions (such as edges), enabling passive salt removal without significantly compromising evaporation performance, the underlying mechanisms governing spatially controlled salt deposition in both 2D and 3D systems remain poorly understood.

Among these, salt crystallization distinguishes itself by synergizing zero-liquid discharge with in situ salt harvesting. A prominent example is Wu’s biomimetic 3D-printed evaporator [22], which employs hierarchical microchannels to regulate interfacial water film homogeneity. The resultant thermal gradient along the evaporator spatially confines salt precipitation to apical regions. The water evaporation rate reached 2.63 kg·m−^2^·h−^1^ under one sun illumination, with a high energy conversion efficiency of 96%. However, this design necessitates complex digital light processing 3D printing protocols, specialized materials, and intricate surface patterning [22], highlighting the unmet need for simple yet precise salt localization control in evaporative desalination.

Most studies focus on structural design but overlook how thermal gradients and water transport affect salt migration pathways in three-dimensional systems. To address this gap, we hypothesize that controlled salt crystallization can be achieved by leveraging surface temperature gradients to modulate water surface tension and, consequently, direct salt transport. Inspired by 3D-cone evaporator architectures, we propose a low-cost conical evaporator fabricated from cellulose paper infused with submicron powdered activated carbon (AC). The composite material combines the broadband solar absorption properties of AC for efficient energy capture with cellulose’s capillary-driven water transport. This facile fabrication method achieves a high evaporation rate of 2.54 kg·m^−2^·h^−1^ and 93.7% efficiency under one sun irradiation. Crucially, the conical geometry establishes a non-uniform temperature gradient, thermodynamically steering salt precipitation to a fixed locus (~1.4 cm from the base) where salt migration and deposition rates equilibrate. By elucidating the underlying phase-transition mechanisms, this work provides a paradigm for spatially controlled salt deposition in solar desalination systems.

## 2. Experimental Section

### 2.1. Materials

Activated carbon (AC) powders and cellulose paper were obtained from the local market. The AC powders were ground into submicron-sized particles using a mortar and pestle. These submicron powders were then washed three times with ultrapure water and ethanol to remove impurities. After washing, the powders were dried overnight in a DHG-9076A oven at 40 °C to eliminate any residual moisture. Sodium chloride (NaCl, AR ≥ 98%) was purchased from Shanghai Aladdin Biochemical Technology Co., Ltd. (Shanghai, China). All chemicals were of analytical grade and used without further purification. Deionized water was used to prepare the necessary solutions and rinse the samples.

### 2.2. Fabrication of Photothermal Evaporators

The process for fabricating the evaporators is illustrated in Figure 1. The submicron AC powders were uniformly mixed with cellulose paper and thoroughly stirred. The mixture was prepared with submicron-activated carbon powder at 75% by weight. The mixture was then poured into conical molds with different height-to-diameter (H/D) ratios (0, 0.25, 0.75, 1, 1.25, 1.75, and 2) with a bottom area of π mm^2^ for molding. To optimize the weight percentage of submicron AC powders in the cellulose paper pulp, we fixed the H/D ratio of the 3D-cone evaporator at 1. For the fabrication process, varying amounts of submicron AC powders (150, 300, 450, and 540 mg) were ultrasonically dispersed in 10 mL of water. Then, 600 mg of paper pulp dispersion was added to the solution and mixed thoroughly with the dispersed AC powders under vigorous stirring for 30 min. Finally, the mixture of submicron AC powders and cellulose paper pulp was poured into conical molds to form the desired shape. The samples were dried overnight in a DHG-9076A oven at 40 °C, resulting in 3D structures featuring micropores for subsequent characterization.

### 2.3. Characterization

Attenuated Total Reflection Fourier Transform Infrared Spectroscopy (ATR-FTIR, Thermo Scientific Nicolet iS10, Thermo Fisher Scientific, Waltham, MA, USA) was used to analyze the surface functional groups of AC and the AC/cellulose composite. The morphology of the samples was observed using a field-emission scanning electron microscope (SEM, JEOL JEM7001F, JEOL Ltd., Akishima, Tokyo, Japan). Optical images of the AC/cellulose were captured with an advanced opto-digital microscope (DSX500, Olympus, Tokyo, Japan). X-ray photoelectron spectroscopy (XPS) measurements were performed using an X-ray photoelectron spectrometer (K-Alpha, Thermo Fisher Scientific, Waltham, MA, USA) with an Al Kα X-ray source (1486.6 eV). Water contact angles were measured using an optical contact angle system (SL200KS, Shanghai Solon Information Technology Co., Ltd., Shanghai, China). The diffuse reflectance spectra of the activated carbon powders were measured using an ultraviolet–visible–near-infrared spectrometer (Shimadzu UV-3600 Plus, Shimadzu Corporation, Tokyo, Japan) coupled with an integrating sphere (ISR-3100, Riken Keiki Co., Ltd., Tokyo, Japan). All infrared images were captured using a Fluke Ti400 (Fluke Corporation, Everett, WA, USA) infrared thermal camera.

### 2.4. Solar-Driven Water Evaporation and Simulated Brine Desalination Tests

The evaporation performance of the 3D-cone evaporator was evaluated under one sun of irradiation (AM1.5G, 1000 W·m^−2^). A Xenon lamp (Solar-300, China Education Au-light Co., Ltd., Beijing, China) served as the simulated solar source. To precisely control the amount of light incident on the evaporators, an aperture was attached during the experiments. For 3D-cone with different H/D ratios, the top projected plane was utilized to normalize the light intensity. Each integrated evaporator system was placed in a 50 mL beaker filled with artificial brine (24.5 wt.% NaCl simulated brine) and exposed to the simulated solar irradiation. The mass change in the beaker was monitored in real time with an electronic balance (accuracy: 0.1 mg) at 1 min intervals, with the decrease in mass reflecting the volume of water evaporated. The tests were conducted at a room temperature of approximately 25 ± 1 °C and a relative humidity of about 40 ± 2% in an open environment. Additionally, the evaporation rate under dark conditions was measured to calibrate the data.

Generally, solar-to-steam efficiency is determined by the fraction of solar energy harnessed to produce steam. In this study, to obtain more realistic and representative results, we accounted for temperature differences at the evaporation surface. The solar-to-steam efficiency (*η*) was calculated using Equations (1)–(3), as outlined in [35,36,37]:(1)$$\eta =\frac{\dot{m}\left(H_{LV}+Q\right)}{P_{in}}$$(2)$$H_{lv}=1.91846\times 10^{6}{\left[\frac{T_{1}}{T_{1}-33.91}\right]}^{2}$$(3)$$Q=c\left(T_{1}-T_{0}\right)$$
where $\dot{m}$ is the net evaporation rate, which was determined by subtracting the dark condition evaporation rate. $H_{lv}$ is the enthalpy of the water phase from liquid to vapor and could be calculated using Equation (2), with $T_{1}$ representing the evaporation temperature between water and air (K). In Equation (3), Q (J·kg^−^^1^) denotes the heat required to raise the water temperature, wherein c is the specific heat of water (4.2 J·g^−^^1^·K^−^^1^), $T_{0}$ is the initial water temperature, and $P_{in}$ is the energy input from the incident light.

### 2.5. Surface Temperature Measurement

Surface temperature distributions were recorded using an infrared thermal camera (Fluke Ti400, thermal sensitivity: <1 °C) at 1 s intervals. The temperature was simultaneously monitored at 1 s intervals using a recorder (SA-JLY01A4, Wuxi Shiao Technology Co., Ltd., Wuxi, China thermal sensitivity: <1.5 °C) to ensure that the temperature deviation between the two monitoring methods did not exceed ±0.5 °C. The camera was positioned perpendicular to the evaporator surface at a fixed distance to avoid angular distortion. Before each test, the ambient background temperature was recorded and input into the Fluke Ti400 device to ensure correct compensation. These procedures collectively ensured consistent and reproducible experimental conditions. All infrared images were captured using a Fluke Ti400 infrared thermal camera. All tests were performed at least in triplicate, and high consistency was observed between repeated measurements.

## 3. Results

This section presents the key results of the study, focusing on the photothermal properties and desalination performance of the 3D-cone evaporator. Particular attention is given to the mechanism of controlled salt crystallization, which is critical for achieving long-term, stable operation under high-salinity conditions.

### 3.1. The Light-to-Heat Conversion Ability of AC/Cellulose

Various proportions of submicron AC powders were compressed into round pills, and their photothermal conversion capabilities were assessed using the thermometric method [10]. Figure 2a shows the absorption spectrum measured by UV–vis–NIR spectrophotometer and represents that in the wavelength range of 200–2500 nm, all photothermal AC/cellulose pills show a relatively low reflectivity (10–15%), which is primarily attributed to the presence of conjugated π bonds in AC powders [38,39]. Under the irradiation of one sun, the measured equilibrium temperatures were 58.1 °C (25% AC), 65.8 °C (50% AC), 67.4 °C (75% AC), and 70.4 °C (90% AC), respectively, as shown in Figure 2b,c. These facts indicate that activated carbon powder has excellent photothermal conversion capability, and the proportion of carbon powder added to cellulose paper increases, leading to enhanced photothermal conversion capability.

To elucidate the photothermal conversion mechanisms of activated carbon (AC) and its cellulose composite, a systematic structural characterization was performed. Figure 3a presents comparative FTIR spectra revealing critical functional group signatures. Both materials exhibit characteristic vibrations at 1011 cm^−^^1^ (C−O−C asymmetric stretching), 1634 cm^−^^1^ (C=O stretching), and 1439 cm^−^^1^ (C=C aromatic stretching) [40]. The cellulose component introduces distinct hydrophilic functionalities, manifested through a broad O−H stretching band at 3436 cm^−^^1^ and weak −CH_2_ asymmetric stretching at 2910 cm^−^^1^. Notably, the C−O−C region (1011 cm^−^^1^) in AC/cellulose resolves into three resolved peaks through spectral deconvolution: 1158 cm^−^^1^ (C−O−C symmetric stretching), 1112 cm^−^^1^ (C−OH stretching), and 1011 cm^−^^1^ (C−O−C asymmetric stretching), collectively confirming enhanced surface hydrophilicity through oxygenated functional groups [39,40,41].

Complementary XPS analysis (Figure 3b,c) provides electronic structure verification. The C1s spectrum of pristine AC deconvolutes into three components: 284.8 eV (C−C/C=C, 64.98%), 286.6 eV (C−O, 25.24%), and 288.7 eV (C=O, 17.78%) [42,43,44]. The AC/cellulose composite maintains these primary features while introducing a new component at 285.9 eV (C−OH, 12.42%), consistent with FTIR observations. Quantitative analysis reveals a significant decrease in the ratio of carbon to oxygen from 2.3 (AC) to 1.7 (AC/cellulose), indicating increased oxygen functionalization. This chemical modification achieves dual functionality: (1) The preserved sp^2^−hybridized carbon network (evidenced by dominant C−C/C=C content) maintains broadband photon harvesting through conjugated π−electron systems [38,45]; and (2) the introduced polar groups (C−O, C=O, C-OH) establish hierarchical water transport channels within the 3D−cone evaporator matrix [39,44]. The synergistic integration of efficient photon capture and rapid water transport establishes the composite as an advanced interfacial evaporation platform.

As depicted in Figure 3d, we illustrate the wetting property of AC, with the contact angle of its surface approximately measuring 79.6° at 0.2 s. Conversely, as shown in Figure 3e, cellulose paper fiber possesses numerous hydrophilic functional groups, exhibiting a high degree of hydrophilicity, with a water contact angle (WCA) close to 0°, as reported in [46,47].

The surface morphologies of AC and AC/cellulose were analyzed. Figure 3f displays optical photographs of the AC/cellulose material, revealing fabric bundles composed of cellulose paper fibers with a width of 25 μm. The upper surface of these bundles is covered with submicron-sized activated carbon powder. Figure 3g presents SEM photographs of the AC/cellulose material, showing that each fiber consists of several longitudinal textured grooves of varying depths that are discontinuous. These grooves, along with the overlapping fabric bundles, ensure a consistent water supply within the evaporator. Additionally, their increased surface area provides pathways for steam release and air penetration, enhancing the overall efficiency of the evaporation process [48].

### 3.2. Solar-Driven Interfacial Evaporation of 3D-Cone Evaporator

To further investigate the solar-driven water evaporation performance, we evaluated the 3D-cone evaporator by floating it on the water surface. As illustrated in Appendix A, for ease of comparison, each 3D-cone evaporator is constructed from cellulose paper with varying weight percentages of activated carbon powder: 0 wt.%, 25 wt.%, 50 wt.%, 75 wt.%, and 90 wt.%. Among these, a 3D-cone evaporator containing 75% activated carbon exhibits the fastest evaporation rate under one sun irradiation, achieving a rate of 2.54 kg·m^−^^2^·h^−^^1^.

Next, we designed three-dimensional evaporators with different height-to-diameter (H/D) ratios, all made of cellulose paper with the addition of 75 wt.% AC powder. For comparison, a 2D plane (with a bottom area of π mm^2^) was also prepared using the same methods. As shown in Figure 4a, the varying H/D ratios are implemented in the 3D-cone evaporators. Figure 4b presents the typical curves of time-dependent mass change for these evaporators, with H/D ratios ranging from 0 to 0.25, 0.75, 1, 1.25, 1.75, and 2. Under one sun irradiation, the average water evaporation rate increased from ~1.76 kg·m^−^^2^·h^−^^1^ to 2.29 kg·m^−^^2^·h^−^^1^, 2.44 kg·m^−^^2^·h^−^^1^, 2.54 kg·m^−^^2^·h^−^^1^, 2.64 kg·m^−^^2^·h^−^^1^, 2.66 kg·m^−^^2^·h^−^^1^, and 2.67 kg·m^−^^2^·h^−^^1^, respectively. A similar trend was observed in darkness, as shown in Appendix A.

To monitor the temperature evolution during the evaporation process, an infrared camera was utilized. As depicted in Figure 4d, both in dark (0 min) and illuminated conditions (before 60 min), as the H/D ratio increases, there is a slight change in the temperature at the bottom of the three-dimensional evaporator, while the temperature at the top shows a downward trend. It suggests that a larger H/D ratio enhances the brine supply within the evaporator, thereby improving the evaporation rate.

As shown in Figure 4c, the solar-to-steam efficiency was determined based on Equations (1)–(3) provided in Section 2.4 with the structure H/D ratio increasing from 0 to 0.25, 0.75, 1, 1.25, 1.75, and 2. The solar-to-heat efficiencies were calculated to be approximately 85.4%, 89.7%, 89.2%, 93.7%, 90.3%, 87.7%, and 91.7%, respectively. The highest solar-to-steam efficiency is achieved with the three-dimensional evaporator at an H/D ratio of one, indicating a balanced position for photothermal water evaporation and water transport, resulting in the fastest net evaporation rate.

### 3.3. Desalination Performance of the 3D-Cone Evaporator

#### 3.3.1. Salt-Resistant Property of the 3D-Cone Evaporator

High salinity, 24.5 wt.%, of sodium chloride (NaCl) solution was employed to assess the salt resistance of the fabricated 3D conical evaporator. This concentration corresponds to the saturation limit of NaCl at room temperature. No other salts were included in the solution. The salt-resistant performance of the developed 3D-cone evaporator has been assessed. We designed a series of 3D-cone evaporators with different height-to-diameter (H/D) ratios and conducted continuous experiments for 8 h using a 24.5 wt.% NaCl solution. The evaporation rates for these evaporators are presented in Appendix A. Intriguingly, as the desalination process progressed, visible salt accumulation was observed only at a specific location (~1.4 cm from the conical base) of the conical evaporator, as shown in Figure 5a. To monitor the temperature evolution during the evaporation process, we utilized an infrared camera and an infrared thermometer. Temperatures at the top, bottom, and middle of the evaporator were selected as representative points for monitoring, and the measurements were averaged over three readings taken at the same location on the 3D-cone evaporator. The temperature curves are depicted in Figure 5b–d.

A 3D-cone evaporator with an H/D ratio of 0.25 under one sun irradiation is shown in Figure 5b. Due to the small H/D ratio, the evaporator can be approximately considered as a 2D plane with brine covering its surface. When an equilibrium state is reached, there is a minimal temperature difference between the top and the bottom. The continuous water evaporation and transport of brine from the bottom to the top result in a negligible salt concentration gradient. This concentration gradient facilitates spontaneous salt exchange between the fiber channel and the substrate through the outer wall of the channel, ultimately causing the salt to redistribute back into the bulk solution [49].

As illustrated in Figure 5c, under one sun irradiation, for the 3D-cone evaporator with an H/D ratio of one, before the light is activated, the temperature increases from the top to the middle and then to the bottom. The larger specific surface area at the top leads to a higher evaporation rate, which removes more heat and results in a relatively lower temperature. As evaporation progresses, the water films allow the light source to dominate and cause a rapid temperature increase at the top. Salt precipitation occurs on the surface after 70 min, and as time progresses to 120 min, the temperature at the top stabilizes without further increase, while continuous moisture supply to the top leads to continued salt crystallization and growth until it falls off. Under one sun irradiation, as shown in Figure 5d, for the three-dimensional evaporator with an H/D ratio of 1.75, the increased upward water transport within the evaporator due to the larger H/D ratio results in a lower temperature at the top [33,36]. This causes the salt that initially crystallized at the top to dissolve back into the solution after 180 min, enabling fixed-point crystallization at approximately 1.4 cm from the conical base at a steady rate.

Figure 5e summarizes the temperature difference between the top and bottom of the conical evaporator at different H/D ratios. Variations in the temperature difference across the evaporator surface may cause changes in the direction and magnitude of internal brine flow. When the temperature difference between the top and bottom of the evaporator reaches approximately 5 °C, the internal salt migration rate and salt accumulation rate reach equilibrium, leading to fixed-point salt crystallization [50,51]. Appendix A presents the variation in salt precipitation time for three-dimensional evaporators with different H/D ratios, providing a strategy for designing processes that align water and salt channels.

This study focuses on a saturated NaCl solution to isolate and study the crystallization behavior of a single salt. However, real seawater contains a variety of ions (e.g., Mg^2+^, Ca^2+^, and SO_4_^2^−) that can significantly affect salt deposition. Thus, our results represent a simplified case and may not fully capture the complexity of natural seawater crystallization.

#### 3.3.2. Desalination Mechanism Analysis of the 3D-Cone Evaporator

During the steam generation cycle, the concentration of salt within the 3D-cone evaporator increases progressively. Simultaneously, salt ions migrate back into the bulk water through the water channels, driven by the concentration gradient. This process continues until a dynamic equilibrium is established between the rates of salt migration and accumulation, leading to salt deposition. To quantitatively describe these competing factors, the following equations are introduced. Firstly, the net mass flux of brine driven by solar energy at the evaporation interface of the evaporator can be determined by multiplying the solar net evaporation rate by the salinity of the seawater. The migration of seawater occurs in three distinct forms: diffusion, advection, and osmosis. The relevant equations can be expressed as follows [19]:(4)$$J_{salt,\;\;\;accu}=\frac{{\dot{m}}_{evap}C_{salt,bulk}}{1-C_{salt,bulk}}$$(5)$$J_{salt,\;\;\;migration}=J_{diffusion}+J_{advection}+J_{osmotic}$$
where $J_{salt,\;accu}$ and $J_{salt,\;migration}$ represent the accumulation or migration rate of salt, $C_{salt,\;bulk}$ is salt mass fraction at bulk water, and ${\dot{m}}_{evap}$ refers to the net mass flux of vapor.

Second, $J_{diffusion}$, $J_{advection},\;$ and $J_{osmotic}$ are the salt flux through the three mechanisms of diffusion, advection, and osmosis, respectively. In the 3D-cone evaporator, the diffusion of salt ions is the predominant method. Each mechanism can be further elaborated with detailed equations. The salt flux through diffusion can be expressed by Equation (6) [22]:(6)$$J_{diffusion}=D_{salt}P\frac{∆C}{a},D_{salt}\sim \frac{r∆\gamma cos\theta }{2\tau }$$
where $D_{salt}$ is the diffusion coefficient of the brine, P is the porosity of the evaporator, ∆C is the salt concentration gradient between the surface and the bottom of the evaporator, a is the tortuosity of the material, and ∆γ is the difference in surface tension. To be more specific, the two factors, pore size (r) and water contact angle (θ), are consistent for all H/D evaporators, while the latter factors, viscosity (τ), surface tension (γ), and viscosity, are determined by the properties of seawater. Therefore, we consider the changes in surface tension (γ) to adjust the salt flux through diffusion [19,22].(7)$$f=\frac{∆\gamma }{L}=\frac{\gamma_{L}-\gamma_{H}}{L}=\frac{d\gamma }{dT}\cdot \frac{∆T}{L}$$

In Equation (7), the capillary force caused by the temperature difference is represented by f, with $\gamma_{L}$ and $\gamma_{H}$ These are the surface tensions of the brine at low and high temperatures, respectively. The direction of the force is from the high-temperature region to the low-temperature region; dγ/dT is the coefficient of surface tension as a function of temperature; ∆T is the temperature difference between the top and bottom of the evaporator; and L is the distance between the two locations. Combining Equations (6) and (7), the $J_{diffusion}$ induced by temperature difference can be expressed as(8)$$J_{diffusion}=\frac{rLcos\theta }{2\tau }P\frac{∆C}{a}\cdot f$$

As can be seen, $J_{diffusion}$ is directly related to *f*. We will specifically analyze the salt flux through diffusion inside evaporators with H/D ratios of 0.25, 1, and 1.75. For the evaporator with an H/D ratio of 0.25, the brine transport in the porous structure in the presence of evaporation consists of diffusion. As shown in Figure 6d, with equal temperatures across all areas, the concentration gradient will lead to the spontaneous salt exchange between the fiber channel and the substrate through the outside of the channel wall, as shown in Figure 6a, and the pathway of brine is indicated by the red arrows. Under sunlight, for the evaporator with H/D = 1, the brine transport in the porous structure in the presence of evaporation consists of diffusion and osmosis. Among them, diffusion mass transport plays a dominant role. During the evaporation process, the temperature at the top is lower than that at the bottom and middle. This is because water evaporation is dominant, carrying away some heat and causing the top temperature to decrease, as shown in Figure 6e. The $∆T=T_{b}-T_{t}=5.2\;{}^{\circ}C>0$, and the capillary force f_1_ caused by the temperature difference points from the bottom to the top, as depicted in Figure 6b, where the pathway of brine is indicated by the red arrows, ensuring the thorough transport of brine until the salt continuously accumulates and grows at the top.

In the evaporator with an H/D ratio of 1.75, the brine transport within the porous structure, in the presence of evaporation, comprises both diffusion and osmosis. Among these mechanisms, diffusion mass transport predominantly governs the process. Throughout the evaporation cycle, once salt formation initiates at the top, the temperature in the middle tends towards equilibrium, establishing a temperature gradient of 3.5 °C between the top and middle sections, as illustrated in Figure 6f. This temperature difference gives rise to two capillary forces: f_1_, driven by the overall temperature difference $∆T=T_{b}-T_{t}=6.2\;{}^{\circ}C>0$ (black line in Figure 6f), and f_2_, a reverse capillary force pointing downwards due to the local temperature gradient $∆T=T_{m}-T_{t}=-3.5\;{}^{\circ}C<0$ (red line in Figure 6f). As depicted in Figure 6c, the pathway of brine is indicated by the red arrows.

It is noteworthy that, in comparison to an H/D ratio of one, osmosis plays a more prominent role in the evaporator with an H/D ratio of 1.75, leading to an increased radial flow velocity of the brine. This phenomenon arises due to the narrowing cross-section and the continuous reduction in space, which results in uneven brine flow and consequently elevated flow velocity. Specifically, the restricted inlet area with a smaller cross-sectional area facilitates a higher replenishment rate of brine. This also accounts for the decrease in temperature in the middle of the evaporator with an H/D ratio of 1.75 and the delayed salt discharge at the top of the evaporator.

The stable distance of water spreading upwards is approximately 1.4 cm, at which point the rates of salt migration and accumulation achieve an equilibrium, leading to localized crystallization at a consistent rate. Appendix A illustrate the salt production process in three-dimensional evaporators with varying H/D ratios. Notably, the distance from the bottom of the evaporator to the bottom of the salt outlet remains constant, regardless of the H/D ratio. Compared to current desalination strategies, as depicted in Appendix A, this innovative approach showcases the potential for large-scale desalination through enhanced water evaporation. Compared to previously reported evaporators, our 3D conical design achieves a higher evaporation rate and extended operational stability under high salinity. For instance, Li [52] reported an evaporation rate of 1.39 kg·m−^2^·h−^1^ with operational stability of 10 h under 3.5 wt.% salinity. In contrast, our system maintains 2.54 kg·m−^2^·h−^1^ for 8 h under 24.5 wt.%, demonstrating significant improvement in both efficiency and salt resistance. While some systems reach comparable evaporation rates (>3 kg·m−^2^·h−^1^), they often rely on complex nanomaterials or external salt removal mechanisms [53,54]. Our use of low-cost cellulose paper provides a simpler, scalable alternative. It presents a sustainable and practical solution to address the global freshwater crisis.

Despite the promising performance, this study has certain limitations. All experiments were conducted under controlled indoor conditions, and the long-term durability of the cellulose-based evaporator in real seawater environments remains to be evaluated. In addition, practical challenges such as large-scale fabrication and environmental adaptability (e.g., wind, fouling) require further investigation in future work. Future research should focus on enhancing material robustness, validating performance in field settings, and developing scalable designs to facilitate practical implementation in sustainable water treatment applications.

## 4. Conclusions

In summary, we have developed a novel conical evaporator utilizing a cellulose paper matrix, which achieves directional salt crystallization control through an optimized conical height-to-diameter structure design. This innovative configuration enables precise localization of salt deposition at a predetermined position (~1.4 cm from the conical base). Rational design of transport pathways proves crucial for suppressing salt crystallization at the evaporation interface and sustaining stable operation over prolonged durations. Through comprehensive semi-quantitative analysis, we have elucidated the underlying salt transport mechanisms during the evaporation process. The evaporator’s superior performance stems from its synergistic material composition and structural engineering, combining the exceptional photothermal conversion properties of activated carbon with the hydrophilic functionality of cellulose. Under one sun irradiation, the system demonstrates remarkable performance metrics, achieving an evaporation rate of 2.54 kg·m^−2^·h^−1^ with 93.7% solar-to-steam efficiency, while maintaining stable operation for 8 h in high-salinity conditions (24.5 wt.%). This cost-effective and scalable approach offers a sustainable solution for advanced solar-driven water purification systems. Looking forward, future work could explore the use of this design in multi-ion or real seawater conditions, long-term anti-fouling performance, and potential integration into modular, off-grid water purification devices for coastal regions.

## Figures and Tables

**Figure 1 materials-18-02610-f001:**
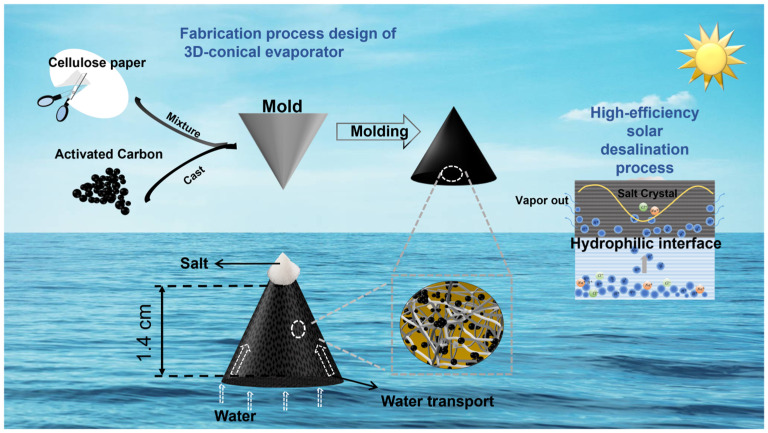
Schematic diagram of preparation of the cellulose paper-based 3D-cone solar evaporator.

**Figure 2 materials-18-02610-f002:**
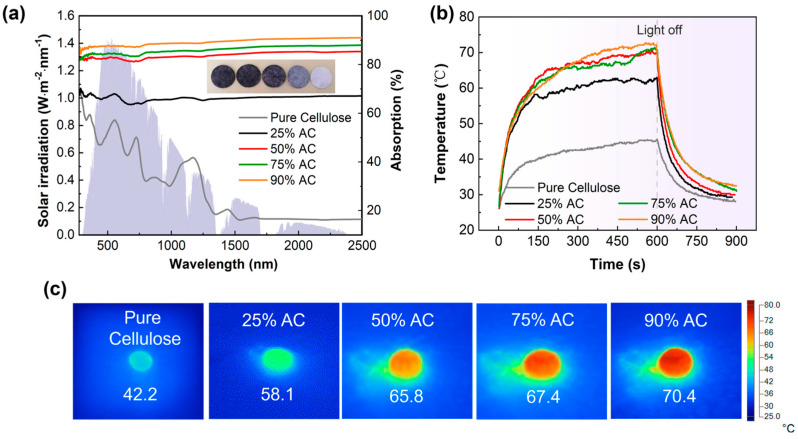
(**a**) Absorption spectra of different AC/cellulose pills. The gray background represents the frequency−dependent solar spectrum (AM1.5G); (**b**) surface temperature variations in different AC/cellulose pills (~2 mm in thickness) under one sun irradiation; (**c**) IR images of pure cellulose, 25% AC, 50% AC, 75% AC, and 90% AC pills under one sun irradiation.

**Figure 3 materials-18-02610-f003:**
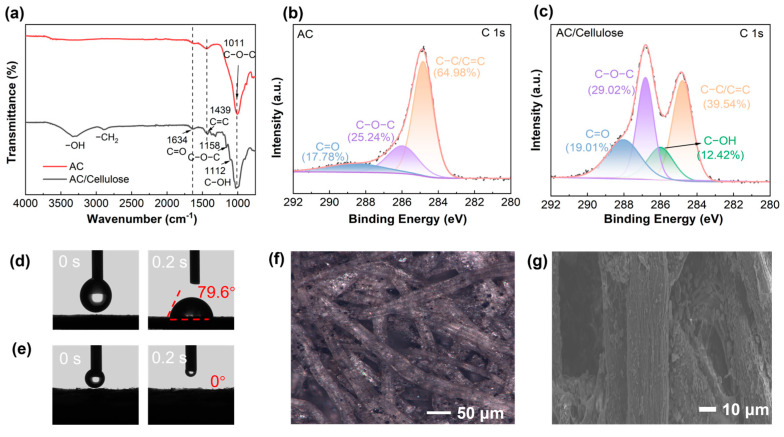
(**a**) FTIR spectra of AC and AC/cellulose; (**b**,**c**) high−resolution XPS spectra of C1s peak of AC powders and AC/cellulose; (**d**,**e**) contact angle test of the AC and AC/cellulose; (**f**) optical digital microscope pictures of AC/cellulose materials; and (**g**) SEM image of AC/cellulose paper surface.

**Figure 4 materials-18-02610-f004:**
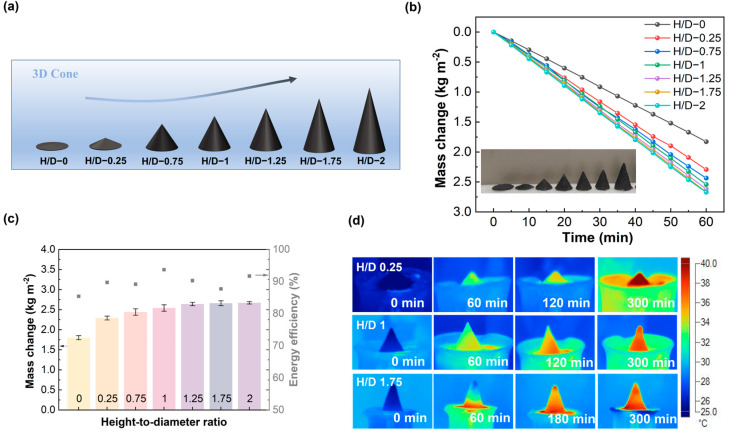
(**a**) Schematic diagram of 3D-cone evaporators with different height-diameter ratios; (**b**) mass change curves of 3D-cone evaporators with different height-diameter ratios under one sun irradiation; (**c**) evaporation rate and the corresponding solar-to-steam efficiency of 3D-cone evaporators with variation in H/D ratios under one sun irradiation; the error bars are the standard deviations of the mean evaporation rates (*n* = 3, *n* is the number of individual observations for each sample); and (**d**) infrared image of 3D-cone evaporator with H/D of 0.25, 1, and 1.75 at different times.

**Figure 5 materials-18-02610-f005:**
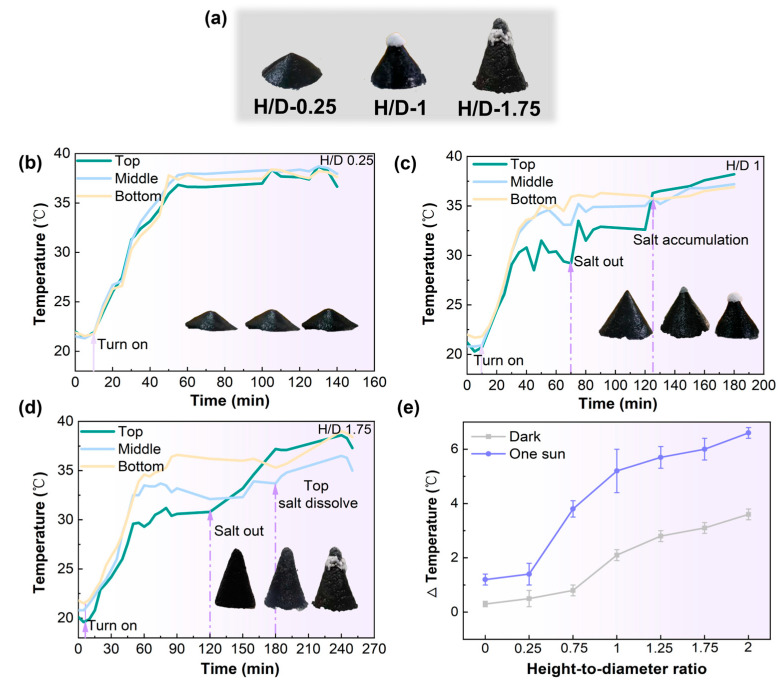
(**a**) The digital photographs of the 3D-cone evaporator with salt precipitation on evaporators with H/D of 0.25, 1, and 1.75; and (**b**–**d**) temperature evolution along with time at different vertical positions on evaporators with H/D of 0.25, 1, and 1.75 under one sun irradiation. The inset shows the localized crystallization process on the 3D-cone evaporator. Data represent the average of three repeated measurements. Error bars are omitted for clarity due to minimal standard deviation. (**e**) Temperature differences between the bottom and the top surface temperature with the variation in H/D ratios under both darkness and one sun irradiation conditions (60 min). The error bars are the standard deviations of the mean temperature (*n* = 3, *n* is the number of individual observations for each sample).

**Figure 6 materials-18-02610-f006:**
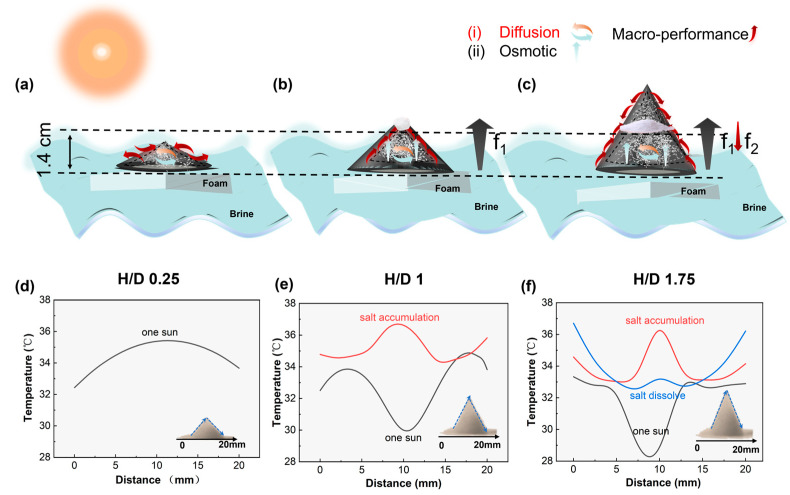
(**a**–**c**) Schematic diagram of 3D-cone: the directions of salt diffusion and permeation (orange and blue arrows) and the pathway for salt discharge (red arrow) with H/D ratios of 0.25, 1, and 1.75, respectively; (**d**) temperature profiles at different horizontal positions of a 3D-cone evaporator with an H/D ratio of 0.25 during sunlight exposure; (**e**) temperature profiles at different horizontal positions of a 3D-cone evaporator with an H/D ratio of one during two stages: sunlight exposure and salt accumulation; and (**f**) the temperature at different horizontal positions of a 3D-cone evaporator with an H/D ratio of 1.75 during three stages: sunlight exposure, salt accumulation, and salt dissolve. Data represent the average of three repeated measurements. Error bars are omitted for clarity due to minimal standard deviation.

## Data Availability

Data will be made available on request.

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
