# Peer review of "Insights into Localized Crystallization in the 3D-Cone Solar Evaporator for High-Salinity Desalination"

_materials, 2025, doi:10.3390/ma18112610_

Round 1
Reviewer 1 Report
Comments and Suggestions for Authors
The current study designed an innovative evaporator to achieve controlled salt crystallization and to mitigate salt precipitation on evaporator surfaces. This evaporator worked continuously for 8 h under 24 wt% salinity conditions showing good performance and a energy conversion efficiency of 93.7%. The design and protocols are sound, and the characterization is rigorous; however, the following comments must be addressed before publication in this journal.
- Introduction section: The authors might want to include some specific data on the current achievements in the literature (i.e., numbers related to evaporation rate, efficiency, etc.). This way, the readers have an accurate idea of the state of the art.
- Section 2.4. According to the title of the section, real seawater is used; however, it is not mentioned in the subsequent paragraph. Please elaborate.
- Section 3.3.1. Line 245, now the authors describe the test solution as a 24.5 wt.% brine solution. Is it seawater or a NaCl solution? Therefore, the composition remains unknown. This is important to predict the type of salt deposition forming.
- The reviewer's main comment is about the composition of the test solution. The main reason is the ions present in this solution. The crystallization of a synthetic NaCl-based solution is different from that of seawater containing a complex matrix of Ca, Mg, SO4, Na, Cl, etc.
- The conclusions section is a mere repetition of the results. The authors might want to add implications of the results and even future perspectives.
- Figure S6, the Ref-axis is difficult to read.
Reviewer 2 Report
Comments and Suggestions for Authors
The authors designed a three-dimensional conical evaporator using low-cost cellulose paper for efficient solar-driven desalination. This evaporator design achieved controlled salt crystallization by meticulously balancing the rates of salt diffusion and accumulation, thereby directing salt precipitation to a predetermined location approximately 1.4 cm above the conical base. The topic of the manuscript is of interest; however, several key improvements are necessary to enhance its quality, clarity, and scientific rigor. Please consider the following specific suggestions.
1.The abstract currently places too much emphasis on background context. Instead, it should clearly highlight the novelties, key contributions, and quantitative findings of the study. Ensure it is concise and focused.
2.Avoid using lumped or grouped references without context. Instead, briefly explain the relevance and key findings of the cited studies to better frame the background.
3.The literature review includes outdated sources (e.g., 2006, 2014, 2015). Please incorporate recent studies (last 2–3 years) to reflect current advancements. For instance, the following paper published in 2025 is related to solar evaporation and desalination which is related to your paper. You can use it for this purpose.
https://www.sciencedirect.com/science/article/pii/S1359431125011238
4.The research gap, hypothesis, novel aspects, and contributions of the work should be explicitly stated in the Introduction or a separate subsection to better define the scope and importance of the study.
5.Clearly explain how the experimental outputs were measured. What instruments were used? What were the measurement intervals and precision?
6.Provide a brief overview of Section 3 before jumping into subsection 3.1 to guide readers through its objectives and structure.
7.Elaborate on how experimental stability was achieved and maintained throughout the testing. Was there any calibration or control mechanism applied?
8.Clarify whether error analysis or uncertainty quantification was conducted. This is crucial for validating the reliability of experimental findings.
9.All relevant equations should be presented within the Materials and Methods section for clarity and completeness.
10.Compare your experimental results with existing technologies or prior studies to highlight advantages, improvements, or trade-offs.
11.Clearly identify the limitations of your study, both in terms of methodology and practical implementation.
12.Conclude with practical recommendations and suggest future research directions to extend and build upon the current work.
Round 2
Reviewer 1 Report
Comments and Suggestions for Authors
The authors have addressed all the comments, as such, this manuscript is suitable for publication.
Reviewer 2 Report
Comments and Suggestions for Authors
No additional comment. Thanks.